# The Effects of Tocotrienol on Gut Microbiota: A Scoping Review

**DOI:** 10.3390/life13091882

**Published:** 2023-09-07

**Authors:** Aswini Kumareswaran, Sophia Ogechi Ekeuku, Norazlina Mohamed, Norliza Muhammad, Alfizah Hanafiah, Kok-Lun Pang, Sok Kuan Wong, Deborah Chia Hsin Chew, Kok-Yong Chin

**Affiliations:** 1Department of Pharmacology, Faculty of Medicine, Universiti Kebangsaan Malaysia, Cheras 56000, Malaysia; aswiniash1101@gmail.com (A.K.); azlina@ppukm.ukm.edu.my (N.M.); norliza_ssp@ppukm.ukm.edu.my (N.M.); sokkuan@ukm.edu.my (S.K.W.); 2Department of Biochemistry, Faculty of Medicine, Universiti Kebangsaan Malaysia, Cheras 56000, Malaysia; virgosapphire2088@yahoo.com; 3Department of Medical Microbiology and Immunology, Faculty of Medicine, Universiti Kebangsaan Malaysia, Cheras 56000, Malaysia; alfizah@ppukm.ukm.edu.my; 4Newcastle University Medicine Malaysia, Iskandar Puteri 79200, Malaysia; kok-lun.pang@newcastle.edu.my; 5Department of Medicine, Faculty of Medicine, Universiti Kebangsaan Malaysia, Cheras 56000, Malaysia; deborah_cch@ukm.edu.my

**Keywords:** anti-inflammatory, gut microbiome, tocotrienol, vitamin E

## Abstract

Gut dysbiosis has been associated with many chronic diseases, such as obesity, inflammatory bowel disease, and cancer. Gut dysbiosis triggers these diseases through the activation of the immune system by the endotoxins produced by gut microbiota, which leads to systemic inflammation. In addition to pre-/pro-/postbiotics, many natural products can restore healthy gut microbiota composition. Tocotrienol, which is a subfamily of vitamin E, has been demonstrated to have such effects. This scoping review presents an overview of the effects of tocotrienol on gut microbiota according to the existing scientific literature. A literature search to identify relevant studies was conducted using PubMed, Scopus, and Web of Science. Only original research articles which aligned with the review’s objective were examined. Six relevant studies investigating the effects of tocotrienol on gut microbiota were included. All of the studies used animal models to demonstrate that tocotrienol altered the gut microbiota composition, but none demonstrated the mechanism by which this occurred. The studies induced diseases known to be associated with gut dysbiosis in rats. Tocotrienol partially restored the gut microbiota compositions of the diseased rats so that they resembled those of the healthy rats. Tocotrienol also demonstrated strong anti-inflammatory effects in these animals. In conclusion, tocotrienol could exert anti-inflammatory effects by suppressing inflammation directly or partially by altering the gut microbiota composition, thus achieving its therapeutic effects.

## 1. Introduction

The human body contains many microorganisms, including bacteria, fungi, viruses, and protozoa, which are known collectively as microbiota [1]. The human gastrointestinal tract houses the largest number of these microorganisms, known as gut microbiota, which exert significant physiological and pathological influences on the host [2]. Bacteria, both commensal and pathogenic, represent the most significant type of gut microbiota [3]. The most dominant phyla of gut bacteria include Bacteroidetes, Firmicutes, Proteobacteria, Fusobacteria, Actinobacteria, and Verrucomicrobia. Notably, Bacteroidetes (gram-negative bacteria) and Firmicutes (gram-positive bacteria) represent 90% of the gut microbiota composition [4,5,6].

The gut microbiota influences the host metabolism through several mechanisms, such as enhancing gut integrity, modifying the intestinal epithelium, defending against pathogens, and modulating the host immune system [7,8]. Previous studies have shown that genetic background, lifestyle habits, health status (e.g., infection and inflammation), the use of xenobiotics (e.g., antibiotics, drugs, and food additives), hygiene, environmental factors, and diet (e.g., high sugar or low fiber) can cause gut dysbiosis [9].

Gut dysbiosis is broadly defined as an imbalance in the gut microbiota that results in decreased numbers and diversity of commensal bacteria and increased pathogenic bacteria [10]. Gut dysbiosis has been associated with many pathological conditions, such as inflammatory bowel disease (IBD), obesity, cardiovascular disease, metabolic syndrome, and cancer [11,12]. Since Firmicutes and Bacteroidetes represent the major phyla in the composition of gut microbiota, the Firmicutes-to-Bacteroidetes ratio (F/B ratio) can be used to determine whether there is gut dysbiosis [13]. An altered F/B ratio has been shown to have significant impacts on human health. For instance, compared with a healthy control group, the F/B ratio has been reported to decrease in patients with IBD, while it increases in patients with obesity [14]. It has been suggested that the F/B ratio influences metabolic potential, energy harvest, and weight regulation [15]. An increased F/B ratio has been associated with obesity [16]. Changes in F/B ratio have also been linked to the development of metabolic disorders, such as type 2 diabetes and non-alcoholic fatty liver disease [17]. Another study reported that the F/B ratio was negatively associated with osteoporosis [18].

Furthermore, gut microbiota generates short-chain fatty acids (SCFAs) such as butyrate by metabolizing starch and dietary fibers through fermentation [19]. SCFAs have important physiological roles because they enhance phagocytosis and chemotaxis, exert anti-inflammatory and anti-microbial effects, and alter gut integrity [20,21]. SCFAs such as butyrate prevent pathogens from entering the blood circulation by increasing the mucus production of the intestinal epithelial cells, the secretion of adenosine monophosphates, and oxygen attainability, and by reducing pro-inflammatory cytokines by inhibiting pro-inflammatory immune cells and activating anti-inflammatory immune cells [22]. SCFAs also enhance the functions of tight junction proteins, which is why tight junction dysfunction occurs during dysbiosis [23]. In gut dysbiosis, the synthesis of SCFAs by commensal bacteria is disrupted, and this loosens the tight junction of the intestinal epithelial barrier [24,25].

Dysbiosis can lead to chronic inflammation due to the increased amounts of intracellular pathogenic bacteria [26]. The pathogenic bacteria in the gut release endotoxins such as lipopolysaccharide (LPS), which activates the intestinal macrophages and causes them to produce pro-inflammatory cytokines (e.g., interleukin (IL)-1, IL-6, IL-8, and tumor necrosis factor-alpha (TNF-α)) [27]. The ensuing inflammation increases the permeability of the intestine. The increased intestinal permeability causes endotoxins such as bacterial LPS to enter the blood circulation, leading to the production of immunosuppressive proteins, and subsequently causing immune dysfunction and chronic diseases [28,29,30]. A previous study showed that there was an increase in mucus-degrading bacteria in IBD patients, and this might be responsible for the damage to the intestinal epithelial barrier [31].

By recognizing that gut microbiota dysbiosis and the inflammation associated with it play a key role in causing pathological conditions such as obesity and IBD, it can be deduced that the use of natural compounds that can partially restore gut microbiota diversity to a balanced state, and which possess anti-inflammatory characteristics, could be useful in treating gut dysbiosis-associated diseases. Tocotrienol, a family of unsaturated forms of vitamin E, have exhibited potent anti-inflammatory activity [32]. Tocotrienol is divided into four specific isomers: α, β, γ, and δ (Figure 1) [33,34]. It can be found in many natural sources, such as palm oil, barley germ, coconut oil, rice bran oil, wheat germ, and annatto, in a variety of compositions [35]. The anti-inflammatory effects of tocotrienol have been well-recognized [36]. A previous study showed that δ-tocotrienol (δTE) supplementation reduced heart and liver inflammation in obese rats [37]. Our group has previously reported the anti-osteoporotic and anti-arthritic properties of tocotrienol [38,39,40,41]. Tocotrienol was also shown to enhance the efficacy of daptomycin against methicillin-resistant *Staphylococcus aureus* in an infected wound model [42].

The anti-inflammatory effects of tocotrienol have been well studied. Tocotrienol suppresses the nuclear factor κB (NF-κB) activation pathway, which is closely associated with inflammation [43]. This leads to the inhibition of the downstream activator protein-1 and the suppression of pro-inflammatory cytokines, such as IL-2a, IL-12, IL-18, IL-6, and TNF-α [44,45]. The expression of cyclo-oxygenase 2 (COX-2), one of the main mediators of inflammation, is also suppressed by tocotrienol [46]. As mentioned above, LPS from gram-negative bacteria stimulates COX-2 expression [47] and activates the NF-κB pathway [48]. Thus, tocotrienol can suppress LPS-induced inflammation. Tocotrienol also exerts anti-inflammatory activity by suppressing the release of nitric oxide (NO) by blocking LPS-stimulated inducible nitric oxide synthase (iNOS) and COX-2 expression without affecting COX-1 [49]. Overall, the anti-inflammatory effects of tocotrienol can be exerted through the inhibition of iNOS, COX-2, and NF-ĸB expression.

Given how gut microbiota dysbiosis plays a vital role in the pathogenesis of various diseases, the influence of tocotrienol on the composition of gut microbiota has gained attention because it may explain the many health beneficial effects of this compound. Tocotrienol is reported to improve gut dysbiosis and the F/B ratio in mice with type 2 diabetes [50]. Therefore, tocotrienol could be a potential treatment for gut dysbiosis. This review aims to summarize the effects of tocotrienol on gut microbiota as reported in the recent literature. We hope that this review will advocate the practical application of tocotrienol in correcting gut dysbiosis and improving health.

## 2. Methodology

This scoping review was developed using the Arksey and O’Malley (2005) framework [51], and it was carried out in accordance with the Preferred Reporting Items for Systematic Reviews and Meta-Analyses Extension for Scoping Reviews [52] (Appendix A). The following steps were implemented: (1) defining the research question; (2) gathering the relevant studies; (3) selecting the studies; (4) extracting the data; (5) reporting the results.

### 2.1. Defining the Research Question

The research question was as follows: what are the effects of tocotrienol on gut microbiota? Since vitamin E isomers are usually present together as a mixture in natural sources, natural tocotrienol mixtures containing tocopherols were considered in this review. Gut microbiota is a broad term which includes various commensal and pathogenic bacteria. Their diversity was emphasized in this review.

### 2.2. Identifying Relevant Studies

A literature search was performed via three electronic databases (PubMed, Scopus, and Web of Science) in April 2023 using the following search string: tocotrienol AND (gut OR intestinal OR gastrointestinal) AND (microbiome OR microbiota). All primary studies examining the impact of tocotrienol-rich fraction (TRF) on the gut microbiota were taken into consideration. Articles without relevant results or primary data (e.g., reviews, letters to the editor, perspectives, books, and book chapters) were not considered. Additionally, due to the preliminary nature of the data they present and their potential to duplicate data from full articles, conference abstracts and proceedings were excluded. Articles not written in English were excluded.

### 2.3. Study Selection

Endnote X9 (Clarivate, London, UK) was used to compile the literature. The search results from the three electronic databases were downloaded. Endnote X9 was used to identify and eliminate duplicate entries, and the list was then reviewed manually. A.K. and S.O.E. reviewed the titles and abstracts of the articles to find pertinent studies. The whole texts of the chosen publications were then acquired and reviewed using the inclusion and exclusion criteria. To identify studies that were overlooked during the search, the reference lists of the included publications were checked. To resolve any disagreements, the opinions of other authors were solicited.

### 2.4. Extracting the Data

A.K. and S.O.E. extracted the pertinent data, such as the names of the researchers, publication years, study designs (disease models utilized, type of tocotrienol, dosages, treatment period), and key findings from the chosen studies using a standardized table.

### 2.5. Collating, Summarising, and Reporting the Results

Instead of synthesizing any specific factors, the scoping review method was used to describe the search results due to the heterogeneity of the studies included and the variables of interest that were revealed. The fundamental goal of a scoping review is to give an overview of a field’s advancements. The study designs, amount of tocotrienol (dose and treatment time) used in each study, disease models, and key findings were therefore summarized and are presented below. Research gaps and the role of tocotrienol in managing chronic diseases by preventing gut dysbiosis were explored.

## 3. Results

### 3.1. Study Selection

The literature search was carried out using three electronic databases and found 44 items (PubMed = 13; Scopus = 19; and Web of Science = 12). After removing 17 duplicates, 27 items were subjected to title and abstract screening. Twenty-one items were excluded for various reasons (not within the scope = 10; not an original research article = 11). Six items were subjected to full-text screening, and all were included in this review. The article selection process is illustrated in Figure 2.

### 3.2. Study Characteristics

In a study using mice with azoxymethane (AOM)/dextran sodium sulphate (DSS)-induced colitis-associated colon cancer, an oral δ/γ-tocotrienol (8/1 ratio) 0.035% (~2.2 μmoles daily) and δ-tocotrienol-13-COOH 0.04% (~2.3 μmoles daily) diet was given for 2 months [53]. Two studies supplemented mice with high-fat diet-induced obesity using 400 or 800 mg/kg annatto tocotrienol for 14 weeks [54,55]. One study induced non-alcoholic fatty liver disease in the mice and supplemented them with a mixture of berberine, tocotrienols, and chlorogenic acid (140 mg/kg diet for 24 weeks) [56]. Another study altered the gut microbiota composition of the male mice using antibiotics and supplemented them with α-, γ-, and δ-tocopherol, as well as γ- and δ-tocotrienol, each at a dose of 75 mg/kg for 17 days [57]. Another study fed severe combined immunodeficient (SCID) mice, which had received human colon cancer cell grafts (HT-29 and HCT-116), with essential turmeric oil–curcumin (ETO–Cur) and tocotrienol-rich fraction (TRF) for 34 days [58].

The diversity of the gut microbiota was analyzed in each study. Firstly, operational taxonomic units (OTUs) were used as a measure of community richness via Quantitative Insights Into Microbial Ecology (QIIME) [56,58] and Quantitative Insights Into Microbial Ecology 2 (QIIME 2) [53,54,55]. Next, a redundancy analysis (RDA) was performed to test whether there were alterations in the gut microbiota compositions of the study groups [54,55]. For further analysis, α-diversity and β-diversity were evaluated. Alpha-diversity assesses the within-group diversity using different metrics, such as Shannon indices [56,58], Faith’s phylogenetic diversity (PD), and Pielou evenness [53,54,55]. Beta-diversity involves the comparison of diversity between groups where a principal coordinate analysis (PCoA) has been performed using unweighted UniFrac, weighted UniFrac, Bray Curtis distance [56,58], and Jaccard matrices [53]. Only one study performed a canonical correspondence analysis (CCA) using PAST3 (PAleontological STatistics) to determine the correlation between the relative abundances (at the species level) of gut microbial communities and the environmental variables, including the treatment given [53].

### 3.3. Study Outcomes

The essential turmeric oil–curcumin–tocotrienol-rich fraction (ETO–Cur–TRF) increased the OTUs, the β-diversity, and the species number diversity in the SCID mice engrafted with human colon cancer cells. This increased microbial diversity indicated the successful prevention of gut dysbiosis. At the phylum level, both the control and ETO–Cur–TRF groups showed increased Bacteroides and Firmicutes, while an increase in Proteobacteria and Actinobacteria could only be observed in the ETO–Cur–TRF group. The gut microbial composition showed that the *Porphymonadaceae*, *Rickenellaceae*, *Lactobacillaeceae*, *Desulphovibrionaceae*, *Enterobacteriaceae*, and *Bifidobacteriaceae* families had increased in number, but that the *Bacteroidaceae* family had decreased in number in the ETO–Cur–TRF group compared with the negative control group. *Clostridium XIVa*, *Lactobacillus*, and *Aliistipes* also increased in number in the ETO–Cur–TRF group. To determine whether the gut microbial changes were reflected in the tumors, DNA was extracted from the tumors of the SCID mice, and quantitative polymerase chain reaction (qPCR) was performed. The results showed increased amounts of *Bifidobacteria*, *Lactobacillus*, and *Clostridium IV* in the ETO–Cur–TRF group. Tumor growth inhibition was more effective in the ETO–Cur–TRF group than in other groups, which received individual supplements [58].

The mice with colitis-associated colon cancer treated with a δ/γ-tocotrienol (8/1 ratio) 0.035% (~2.2 μM) or δ-tocotrienol-COOH 0.04% (~2.3 μM) diet showed decreases in their F/B ratios [53]. This change was associated with decreased tumor size (for both the δ-tocotrienol-COOH- and the δ/γ-tocotrienol-supplemented groups) and tumor formation rate (for the tocotrienol-COOH-supplemented group) [53]. The 800 mg/kg annatto tocotrienol diet significantly increased the F/B ratios of the high-fat diet-fed mice [55]. Concurrently, the mice demonstrated a decrease in fat pad weight, circulating glucose, and adipokine and IL-6 levels [55]. Another study, in which obese mice were treated with annatto tocotrienol (400 mg/kg diet) and green tea polyphenols (GTP, 0.5% w/v in drinking water), only found a decrease in the Firmicutes phyla in the mice treated with green tea polyphenols and those which received the combined treatment [54]. The same study found an improvement in skeletal parameters and a reduction in white adipose tissue in the groups which received annatto tocotrienol and GTP, both individually and in combination. However, the skeletal effects were mostly a result of the GTP [54]. The F/B ratio was partially restored in a non-alcoholic fatty liver disease mouse model which was given a plant extract consisting of 5.27 mg of tocotrienols along with 87.84 mg of berberine and 5.28 mg of chlorogenic acid as part of a 140 mg/kg diet [56]. The plant extract reduced their fasting blood glucose and prevented hyperinsulinemia, though it did not prevent liver steatosis [56]. However, since a mixture of compounds was used in this study, the effects of tocotrienol alone could not be determined.

Microbiota richness, represented by α-diversity, was not significantly altered in any of the studies after supplementation with tocotrienol alone or mixtures containing tocotrienol [53,54,55,56,58]. The studies that used tocotrienol alone were scrutinized to determine the specific bacterial population modified by tocotrienol. In one study, annatto tocotrienol reduced the amounts of Firmicutes as well as the amounts of bacteria in the *Ruminococcus lactaris*, *Dorea longicatena*, and *Lachnospiraceae* families in mice fed with a high-fat diet [55]. Delta-tocotrienol has been shown to increase the amounts of the *Streptococccaceae*, *Bacteroides*, and *Lactococcus* bacteria, as well as the amount of *Parabacteroides goldsteinii CL02T12C30* in the mice with colon cancer who received the supplement compared with those that did not. A significant increase in *Eubacterium coprostanoligenes* and a decrease in the *Clostridiales vadinBB60* group was observed in the mice who received the δ-tocotrienol supplement, but not in the mice who received the δ-T3-13-COOH supplement (a metabolite of δ-tocotrienol). It is still unclear how δ-T3-13-COOH and δ-tocotrienol interact with gut microbiota, so there is no valid explanation for this discrepancy. Furthermore, additional studies are required to determine whether these compounds can modulate gut microbiota in a non-disease model [53].

On the other hand, another study reported that the administration of antibiotics depleted the gut commensal bacteria, but increased the bioavailability of δ- and γ-tocotrienol by 150 and 157%, respectively. This was further evidenced by the lack of tocotrienol or tocopherol in the fecal and urine samples [57]. These observations show that the presence of tocotrienol-metabolizing gut microbiota reduces the bioavailability of tocotrienol.

A summary of the effects of tocotrienol on gut microbiota is presented in Table 1.

## 4. Discussion

Both commensal and pathogenic bacteria reside in our gut. The commensal bacteria regulate the mucosal immune system while the pathogenic bacteria cause immune dysfunction [59]. The disturbance of the composition of the gut microbiota can lead to disease. For instance, apart from the well-recognized F/B ratio, an increase in the abundance of Proteobacteria and facultative bacteria of the *Enterobacteriaceae* family is associated with dysbiosis as their numbers are minimal in healthy humans [60]. Commensal bacteria, which come, for example, from the Firmicutes phylum, maintain the intestinal epithelial barrier by producing SCFAs, such as butyrate [61]. In gut dysbiosis, SCFA-producing bacteria are depleted [62] and mucus-degrading bacteria increase [31], thus increasing intestinal permeability due to intestinal epithelial barrier dysfunction [63]. The dysfunction provides an opportunity for LPS from pathogenic bacteria to enter the blood circulation [64]. The LPS then triggers the immune response by interacting with CD4 membrane receptors to generate proinflammatory cytokines such as IL-6, IL-1, and TNF-α [65]. According to a previous study, LPS-induced inflammation is closely related to type 2 diabetes, cardiovascular diseases, and non-alcoholic steatohepatitis [66]. Tocotrienol could counteract dysbiosis and its associated immune dysregulation. Tocotrienol can decrease the F/B ratio in high-fat diet mice [55] and in colitis-associated colon cancer mice [53]. In another study, tocotrienol with essential turmeric oil increased the beta diversity and the species number diversity in SCID mice. This increase in microbiota diversity is further supported by a significant increase in the phyla of Bacteroidetes, Firmicutes, Proteobacteria, and Actinobacteria [58]. Additionally, tocotrienol increases the abundance of Firmicutes and Bacteroidetes phyla in mice with type 2 diabetes [50].

The alteration of gut microbiota has been associated with many chronic diseases. Recent studies have shown that the F/B ratio increases in overweight or obese patients [67] and decreases in IBD patients [68]. In the current review, gut dysbiosis has been established in several mouse disease models, such as colitis-associated colon cancer [53], obesity [54,55], and non-alcoholic fatty liver disease [56]. Tocotrienol has been shown to prevent or reverse these diseases by correcting the associated gut dysbiosis. The included studies show that parts of tocotrienol’s mechanisms of action can be attributed to its effects on gut microbiota. In mice with AOM/DSS-induced colon cancer, oral supplementation with a δ/γ-tocotrienol 0.035% (~2.2 μmoles daily) and δ-tocotrienol-13-COOH 0.04% (~2.3 μmoles daily) diet led to a decrease in pro-inflammatory cytokines, such as GM-CSF and MCP-1, in the δ-tocotrienol-13-COOH-treated group, and a decrease in IL-1β in the δ-T3-treated group compared with the negative control group [53]. It should be noted that tocotrienol is an anti-inflammatory agent by itself [32]. Unless the gut microbiota is suppressed, it is difficult to determine whether the suppression of inflammation is due to the direct action of tocotrienol or whether it occurs indirectly through the gut microbiota.

The relationship between gut microbiota and tocotrienol is two-way. Although tocotrienol can alter the composition of gut microbiota, gut microbiota also metabolizes tocotrienol. A study has demonstrated that the presence of gut microbiota decreases the bioavailability of tocotrienol because tocotrienol is partially metabolized by the gut microbiota [57]. Since gut microbiota also degrade the side-chain of vitamin E, which subsequently decreases its bioavailability, it would be useful to discover ways of enhancing the bioavailability of tocotrienol without affecting the balance of the gut microbiota. Tocotrienol suffers from low bioavailability because the presence of α-tocopherol transfer protein in the liver facilitates the transport of α-tocopherol (rather than tocotrienol) into the circulatory system [69,70]. Currently, emulsification is a popular method of increasing the bioavailability of tocotrienol, although some components, such as synthetic surfactants, may cause gut irritation [71]. Further studies on the bioavailability of vitamin E in the intestines will be needed to study the effects of intestinal microbiota on vitamin E [57].

The anti-inflammatory actions of tocotrienol have been well-recognized [36]. Our previous study showed that annatto tocotrienol (60 and 100 mg/kg/day for 12 weeks) reduced circulating IL-1α and IL-6 levels in male rats fed with a high-fat, high-carbohydrate diet [72]. It has also been shown to reduce liver inflammation by suppressing toll-like receptor activation and increasing the expression of IL-10 (an anti-inflammatory cytokine) in the same groups of rats [73]. However, it is difficult to determine whether the anti-inflammatory effects of tocotrienol observed in these studies are mediated by alterations in the composition of gut microbiota. This is because tocotrienol has direct immuno-modulatory effects. Recent studies have shown that α-tocotrienol is involved in Th17 differentiation in vitro and in vivo through the IL-6/Janus kinase/STAT3 pathway [74]. On the other hand, γ-tocotrienol has been shown to raise circulating CD4+/CD8+ T-cells and natural killer cells, but to suppress regulatory T-cells in mice with mammary cancer. CD4+ T-cells were found to have infiltrated the tumors of γ-tocotrienol-fed mice [75]. These studies have shown that the immuno-modulatory effects of γ-tocotrienol could be dependent on disease status.

The mechanisms by which tocotrienol acts on microorganisms are not clearly understood. The mevalonate pathway, which is governed by the rate-limiting enzyme 3-hydroxy-3-methylglutaryl-CoA reductase (HMGR), is critical in synthesizing isopentenyl diphosphate for peptidoglycan cell walls, as well as ubiquinones and menaquinones for the electron transport chain, especially in G+C gram-positive bacteria [76]. This pathway may potentially be affected by tocotrienol since studies have shown that tocotrienol suppresses mammalian HMGR expression at the post-translational level [77]. However, its efficacy in suppressing bacterial HMGR is yet to be validated. Efflux pumps form an important bacterial multidrug resistance mechanism [78]. Alpha-tocopherol, the more common form of vitamin E, has been shown to inhibit the efflux pumps of methicillin-resistant *Staphylococcus aureus* [79]. Currently, there is no direct evidence showing that tocotrienol can inhibit bacterial efflux pumps, but it has been shown to suppress the P-gp protein and mdr1 mRNA in breast cancer cells [80]. Overall, the microbial modulating mechanism of tocotrienol remains speculative at the moment and warrants further investigation.

Furthermore, the disease models that have been used to study the effect of tocotrienol on gut microbiota are very limited. Similar studies should be undertaken, but with other disease models. Gut microbiota plays a significant role in diseases such as type 2 diabetes, osteoporosis, and cardiovascular diseases [81,82,83], wherein tocotrienol has demonstrated protective effects. It will be interesting to see whether the protective effects of tocotrienol against these diseases are modulated by gut microbiota. Although rodent models are commonly used in gut microbiota research because they share some physiological and anatomical characteristics with humans (particularly the gastrointestinal system), some variations between the two organisms have been observed. For example, humans have a small caecum in which no fermentation occurs, while mice have a large caecum in which prominent fermentation activities occur [84]. These differences should be considered during experimental design and interpretation. The phylogenetic makeup of the bacteria in both humans and rodents is similar: the two main phyla found in the intestinal tract are Firmicutes and Bacteroidetes [85]. Previous studies have also found that rodents and humans share around 80 microbial genera. However, there are variations in genera; some, for instance, have been found in humans, but not in rodents [86]. Therefore, a clinical trial would be necessary to ensure the translatability of rodent data to humans. A search was performed in the ClinicalTrials.gov registry using the search string specified in Section 2.2 in May 2023. Two clinical trials investigating the effects of annatto tocotrienol on gut microbiota in postmenopausal women with obesity (identifier: NCT03705845; status: recruiting) and sarcopenia (identifier: NCT03708354; status: withdrawn due to difficulties in recruiting) were found. Two additional studies investigating the effects of tocotrienol and carotene-rich red palm oil on inflammation and gut health in adults (identifier: NCT05791370; status: completed, but report not found) and the effects of red palm oil-containing biscuits on gut microbiota in children with vitamin A deficiency (identifier: NCT03256123; status: unknown, and report not found) were also found. The publication of the findings from these studies will help us to understand the effects of tocotrienol on human gut microbiota in the future.

In this review, the literature search was performed using three electronic databases. Since unpublished materials and grey literature were not searched, potential studies and, in particular, studies which produced negative results, could have been missed. A limited number of studies was retrieved from the search despite the use of a broad search string, indicating that more intensive research on this topic is necessary.

## 5. Conclusions

In conclusion, preclinical studies have shown that tocotrienol can restore gut microbiota diversity in various disease models. Gut dysbiosis plays a vital role in the pathogenesis of many diseases, and tocotrienol has been shown to alter the abundance of certain bacterial species depending on the disease model. However, the mechanism by which tocotrienol modulates gut microbiota remains elusive. The improvements in health status observed in the various disease models summarized in this review could be due to the restoration of gut microbiota composition, and the suppression of inflammation could have resulted from the correction of gut dysbiosis or the direct action of tocotrienol. Due to the limited evidence from the clinical trials, it cannot be confirmed whether tocotrienol-induced gut microbial diversity changes can be achieved in patients with disease associated with gut dysbiosis. More research should be carried out to validate whether tocotrienol can be used to manage gut dysbiosis, which is responsible for the pathogenesis of many diseases.

## Figures and Tables

**Figure 1 life-13-01882-f001:**
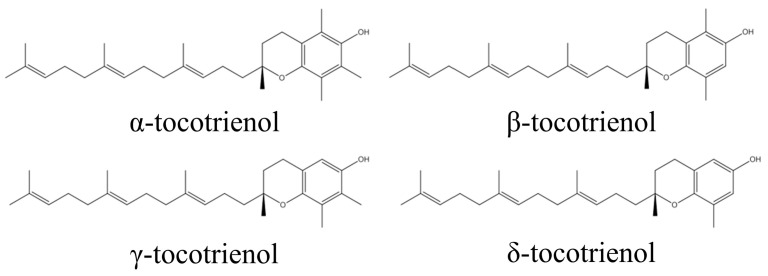
Molecular structure of tocotrienol.

**Figure 2 life-13-01882-f002:**
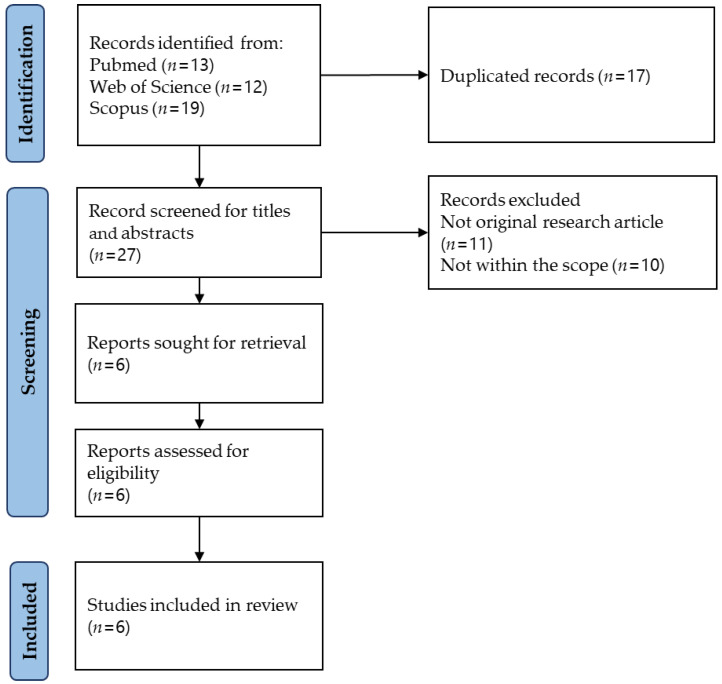
Article selection process.

**Table 1 life-13-01882-t001:** Effects of tocotrienol on gut microbiota.

Researchers	Study Design	Findings
Yang et al., 2021 [53]	Animals: male BALB/c mice (6–7 weeks old) Disease model: colitis-associated colon cancer induced via i.p. injection with AOM at 9.5 mg/kg body weight and 1.5% DSS in drinking water for 1 week. Treatment: δ/γ-tocotrienol (T3) (8/1) 0.035% (~2.2 μM daily) and δ-T3-13-COOH 0.04% (~2.3 μM daily) diet; 2 months Control diet: AIN-93G diet	Gut microbiota composition: Firmicutes-to-Bacteroidetes ratio ↓ δ-T3 and δ-T3-13-COOH vs. AIN-93G diet Family ↑ *Streptococccaceae* in δ-T3 and δ-T3-13-COOH vs. AIN-93G diet ↑ *Eubacterium coprostanoligenes* in δ-T3 vs. δ-T3-13-COOH and AIN-93G ↓ *Clostridiales vadinBB60* group in δT3 vs. δ-T3-13-COOH and AIN-93G Genus ANCOM analysis δ-T3-13-COOH partially ↑ in genus *Roseburia* in AOM/DSS ↑ *Lactococcus* species in δ-T3 and δ-T3-13-COOH vs. AIN-93G LEfSe analysis ↑ *Parabacteroides goldsteinii CL02T12C30* and *Bacteroides* in δ-T3 and δ-T3-13-COOH vs. AIN-93G diet Canonical correspondence analysis (CCA) δ-T3 and δ-T3-13-COOH changed the gut microbiota composition Pro-inflammatory cytokines: ↑ granulocyte-macrophage colony-stimulating factor (GM-CSF), monocyte chemoattractant protein-1 (MCP-1), interleukin 6 (IL-6), tumor necrosis factor-alpha (TNF-α) and Interleukin-1 beta (IL-1β) in AOM/DSS vs. AIN-93G ↓ GM-CSF and MCP-1 in δT3-13-COOH vs. AIN-93G ↓ IL-1β in δ-T3 vs. AIN-93G Health effects: δ-T3-13-COOH reduced total and large-size tumors, δ-T3 only reduced tumor size
Elmassry et al., 2020 [54]	Animals: male C57BL/6J mice (5 weeks old) Disease model: high-fat diet (HFD) in obese mice Treatment: annatto tocotrienol (AT) treatment (400 mg/kg diet) and Green tea polyphenols treatment (0.5% (w/v) GTP in drinking water) for 14 weeks	Gut microbiome composition: Phylum level GTP and GTP+AT: ↓ Firmicutes *A. muciniphila* and *Clostridium* species (in the Clostridiaceae family) ↑ in GTP vs. negative control group *Clostridium symbiosum*, *Dorea longicatena*, *Ruminococcus lactaris*, and *Sporobacter termitidis* ↓ in GTP vs. negative control group *R. lactaris* ↑ in GTP vs. negative control group *Clostridium* (in the Clostridiaceae family), *Clostridum saccharogumia,* and *Subdoligranulum variabile* ↑ in GTP+AT vs. negative control group *Clostridium* (in the Lachnospiraceae family), *C. symbiosum*, *Defluviitalea saccharophila*, *Ruminococcus lactaris*, and *Sporobacter termitidis* ↓ in GTP+AT vs. negative control group Functional profile of the gut microbiome ⟷ in AT vs. negative control group Glycerol degradation, toluene, D-glucarate, allantoin, D-galactarate, and catechol ↓ in GTP vs. negative control Vitamin K2 and urate biosynthesis/inosine-5′-phosphate degradation ↑ in GTP+AT vs. negative control group Amino acid, glycogen, starch, and mannan degradation, and inosine-5′-phosphate biosynthesis ↓ in GTP+AT vs. negative control group Health effects: AT and GTP individually and in combination improved bone health and reduced white adipose tissue. Skeletal effects of GTP > those of AT.
Cossiga et al., 2021 [56]	Animals: male C57BL/6J mice (4 weeks old) Disease model: HFD in a mouse model of non-alcoholic fatty liver disease (NAFLD) Treatment: HFD enriched with plant extracts (HFD+E) (140 mg/kg diet) with 87.84 mg of berberine, 5.27 mg of tocotrienols, and 5.28 mg of chlorogenic acid for 24 weeks. Normal control: standard diet Negative control: HFD	Gut microbiome composition: ⟷ in HFD vs. standard diet ⟷ in HFD+E vs. HFD and standard diet Phylum level Actinobacteria and Firmicutes ↑ in HFD vs. HFD+E and standard diet Bacteroidetes ↑ in HFD+E and standard diet vs. HFD Deferribacteres ↑ in HFD+E vs. standard diet and HFD Verrucomicrobia ↓ in HFD+E vs. standard diet and HFD Firmicutes-to-Bacteroidetes ratio ↓ in standard diet vs. HFD Restored partially in HFD+E Genus level Bacteroides ↑ in HFD+E vs. standard diet and HFD Health effects: The plant extract reduced fasting blood glucose and prevented hyperinsulinemia, though it did not prevent liver steatosis
Ran et al., 2019 [57]	Animals: male C57BL/6J mice (22 g; 7 weeks old) Disease model: alteration of gut microbiota through administration of antibiotics in drinking water; ampicillin (1 mg), sulfamethoxazole (1.6 mg), and trimethoprim (0.32 mg) for 12 days Treatment: mice were intragastrically (i.g.) administered a mixture of vitamin E (mVE, containing α-T, γ-T, δ-T, γ-T3, and δ-T3, each at a dose of 75 mg/kg) in 0.1 mL corn oil daily for 17 days Groups: A groups A1: + antibiotics + vitamin E A2: − antibiotics − vitamin E C groups C1: − antibiotics + vitamin E C2: − antibiotics − vitamin E	Faecal genomic DNA: ↓ in A groups (1–6 ng) vs. C groups (80–125 ng) Not sufficient for 16S rRNA sequencing analysis The antibiotics depleted the gut commensal bacteria Effect of antibiotics on blood levels of tocopherols, tocotrienols, and their metabolites: α-T ⟷ in C1 vs. C2 ↑ in A1 vs. C1 (by 40%) ⟷ in A2 vs. C2 δ-T and γ-T ↑in A1 vs. C1 (by 125%) ⟷ in A2 vs. C2 δ-T3 and γ-T3 ↑ in A1 vs. C1 (by 150–157%) ⟷ in A2 vs. C2 Antibiotics treatment ↑ bioavailability of newly administered VE. Serum delta-carboxyethyl hydroxychroman (δ-CEHC), gamma-carboxyethyl hydroxychroman (γ-CEHC), and alpha-carboxyethyl hydroxychroman (α-CEHC) ↑ in C groups vs. A groups Serum delta-carboxylmethylbutyl hydroxychroman (δ-CMBHC) and gamma-carboxylmethylbutyl hydroxychroman (γ-CMBHC) ↑ in C groups vs. A groups Serum α-CMBHC ⟷ in A groups and C groups Serum levels of longer side-chain degradation metabolites of δ- and γ-forms of VE ↑ in A groups vs. C groups ↓ in long-chain vs. short-chain Liver levels of tocopherols, tocotrienols, and their metabolites Same as serum levels but ↑ in α-T and γ-T3 (3–5 times) vs. serum levels ↑ in A for mVE (by 80–100%), except for α-T CEHC and CMBHC ↓ in A groups vs. C groups, except α-CEHC ↑ γ-CEHC, γ-CMBHC, and α-CMBHC (6 times) vs. serum levels. ↓ in α-CEHC ↑ α-CMBHC in liver vs. α-CMBHC in serum levels in A groups Kidney levels of tocopherols, tocotrienols, and their metabolites Not affected by treatment ↓ in δ-T and γ-T, ½ of γ-T3 but 2(δ-T3) C1 kidney vs. C1 liver CEHC ↓ in A groups vs. C groups, except low levels of α-CEHC α-CEHC ↓ in kidney and liver samples in C1 and C2 vs. α-CMBHC Urine levels of tocopherols, tocotrienols, and their metabolites Tocopherols and tocotrienols were not detected ↑ VE metabolites in C groups vs. blood and tissue levels δ- and γ-forms of metabolites ↑ in the first 12 days but ↓ after antibiotic treatment α-CEHC and CMBHC ↑ after day 10 Faecal levels of tocopherols, tocotrienols, and their metabolites: ↓ 5 VE forms following antibiotic treatment VE metabolites ↑ in first 12 days but ↓ after antibiotic treatment α-CMBHC completely blocked by antibiotic treatment α-CEHC not detected Long-chain metabolites ↓ from 13 to 9 carbons following antibiotic treatment
Chung et al., 2020 [55]	Animals: male C57BL/6J mice (5 weeks old) Disease model: HFD-fed mice Treatment: Annatto-extracted tocotrienol (800 mg/kg diet) (AT) and metformin (200 mg/kg diet) (MET) for 14 weeks Normal control: LFD Negative control: HFD	Gut microbiota profile: RDA Significant association between dietary treatment and variation in gut microbiome Alpha diversity ⟷ LFD, HFD, AT, and MET Most abundant phyla: Firmicutes, Verrucomicrobia, Bacteroides, and Actinobacteria GM in HFD mice Firmicutes-to-Bacteroidetes ratio ↑ in HFD vs. LFD ↓ in AT vs. HFD *Ruminococcus lactaris* and *Alistipes massiliensis* ↑ in HFD vs. LFD *Bifidobacterium bifidum*, *Clostridium disporicum*, *Barnesiella*, *Allobaculum*, and *rc4 -4* ↓ in HFD vs. LFD GM in AT and MET mice *Lachnospiraceae* family ↓ in MET vs. HFD Firmicutes and *D. longicatena* ↓ in AT vs. MET Firmicutes ↓ in AT vs. LFD, HFD, and MET *Verrucomicrobia* ↑ in AT vs. LFD *Ruminococcus lactaris*, *Dorea longicatena*, and *Lachnospiraceae* families ↓ in AT vs. HFD Health effects: AT decreased resistin, leptin, IL-6, and glucose, but did not affect fat pad weight
Farhana et al., 2020 [58]	Disease model: human colon cancer cells HT-29 and HCT-116; HCT-116 cells xenografted into SCID mice Treatment: essential turmeric oil + curcumin (ETO–Cur), tocotrienol-rich fraction (TRF), and ETO–Cur–TRF for 34 days Negative control: untreated tumor-grafted SCID mice	Microbial profiling: Diversity index ⟷ between negative control and ETO–Cur–TRF group Operational taxonomic unit (OTU) ↑ in ETO–Cur–TRF vs. negative control group Beta diversity in phylogenetic tree ↑ in ETO–Cur–TRF vs. negative control group Diversity of species number (n) ↑ in ETO–Cur–TRF vs. negative control group Phylum ↑ Proteobacteria and Actinobacteria in ETO–Cur–TRF group vs. negative control group. Tenericute was eliminated in ETO–Cur–TRF group Composition of microbial family ↑ *Porphymonadaceae*, *Rickenellaceae*, *Lactobacillaeceae*, *Desulphovibrionaceae*, *Enterobacteriaceae*, and *Bifidobacteriaceae* in ETO–Cur–TRF-treated group vs. negative control group ↓ *Bacteroidaceae* in ETO–Cur–TRF group vs. negative control group ↓ *Bacteroides* and *Parabacteroides* in ETO–Cur–TRF group vs. negative control group ↑ *Clostridium XIVa*, *Lactobacillus*, and *Aliistipes* in ETO–Cur–TRF group vs. negative control group ↓ *Bacteroides uniformis* in ETO–Cur–TRF group vs. negative control group Gut microbial changes in tumor ↑ *Bifidobacteria*, *Lactobacillus*, and *Clostridium IV* in ETO–Cur–TRF treated mice vs. negative control group Health effects: ETO–Cur–TRF was more effective than ETO–Cur and TRF in inhibiting cancer growth in vitro and in vivo

Abbreviations: ↑: increase or upregulate; ↓: decrease or downregulate; ⟷: no change or difference; A: antibiotic; AOM: azoxymethane; AT: annatto tocotrienol; C: control; DSS: dextran sulfate sodium; GM: gut microbiota; LFD: low-fat diet; HFD: high-fat diet; MET: metformin; SCID: severe combined immunodeficient mice; α-T: α-tocopherol; δ-T: δ-tocopherol; γ-T: γ-tocopherol; δ-T3: δ-tocotrienol; γ-T3: γ-tocotrienol; VE: vitamin E; GM-CSF: granulocyte-macrophage colony-stimulating factor; MCP-1: monocyte chemoattractant protein-1; IL-6: interleukin 6; TNF-α: tumor necrosis factor-alpha; IL-1β: interleukin-1 beta; GTP: green tea polyphenols; GTP+AT: green tea polyphenols and annatto tocotrienol; NAFLD: non-alcoholic fatty liver disease; δ-CEHC: delta-carboxyethyl hydroxychroman; γ-CEHC: gamma-carboxyethyl hydroxychroman; α-CEHC: alpha-carboxyethyl hydroxychroman; δ-CMBHC: delta-carboxylmethylbutyl hydroxychroman; γ-CMBHC: gamma-carboxylmethylbutyl hydroxychroman.

## Data Availability

No applicable.

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
