# Peer review of "The Effects of Tocotrienol on Gut Microbiota: A Scoping Review"

_life, 2023, doi:10.3390/life13091882_

Round 1

Reviewer 1 Report

Gut dysbiosis is linked to various chronic diseases like obesity, inflammatory bowel disease, and cancer. The activation of the immune system by endotoxins produced by gut microbiota leads to systemic inflammation, contributing to these conditions. Tocotrienol, a subfamily of vitamin E, has shown promising effects in restoring a healthy gut microbiota composition beyond traditional pre-/pro-/postbiotics. This review summarizes existing literature on tocotrienol's impact on gut microbiota. Six relevant studies with animal models demonstrated that tocotrienol altered gut microbiota composition and exhibited strong anti-inflammatory effects. Although this review might be interesting for the communities, only 6 gut microbiota-associated literatures are the major limitation of this review. It would be good to include some immune papers that have been reported by Tocotrienol if possible, since the gut microbiota may influence the immune response and vice versa.

Major comments

Table 1 needs to be more concise, as it summarized the effect of Tocotrienol on gut microbiota. Histology, body weight, serum P1NP level, etc are not associated with the main topic of this Table. More information about alpha diversity should be presented in this part.

The effect of Tocotrienol on immune responses would be better summarized along with the gut microbiota section.

The basic physiology information of tocotrienol should be presented in this review to help the reader know its function.

Author Response

Thank you for reviewing our manuscript. We appreciate your comments, and they are addressed in the attached response sheet.

Reviewer 2 Report

1. Please elaborate on the link between high and low F/B ratios and diseases, or further elaborate on the advantages and disadvantages of high and low ratios on the balance of intestinal microbiota.

2. Please give some examples of what specific microbiota tocotrienols are associated with and what kind of association exists.

3. Whether there is a recovery or improvement in the associated diseases after the intestinal microbiome has been regulated, and whether there is a link between this improvement and tocotrienols.

No glaring deficiencies in wording or grammar were identified。

Author Response

(The authors gave the same response as above.)

Round 2

Reviewer 1 Report

The authors have addressed all my concerns.

Author Response

Thank you. There is no comment to be addressed.